# Gypsum Composites with Modified Waste Expanded Polystyrene

**Pauls P. Argalis \***, **Girts Bumanis** and **Diana Bajare**

Institute of Materials and Structures, Faculty of Civil Engineering, Riga Technical University, Kipsalas Street 6A, LV-1048 Riga, Latvia; girts.bumanis@rtu.lv (G.B.); diana.bajare@rtu.lv (D.B.)
\* Correspondence: pauls-pavils.argalis@rtu.lv

**Abstract:** The construction and demolition waste recycling into secondary raw materials is vital to achieving a sustainable and circular building life cycle. Expanded polystyrene (EPS) is one of the materials whose recycling rate should be increased. EPS boards can be shredded and used as aggregate of lightweight cement composites resulting in a material with combined properties subjected from EPS and mineral binder. To reduce the open structure of shredded EPS particles, proper treatment could improve EPS performance. The heat treatment of the aggregates can reduce the volume and increase their density. In this paper, EPS aggregates were heat-treated at 120 and 130 °C, and heat-modified EPS aggregates with a bulk density of 40 and 100 kg/m$^3$ were incorporated as filler material in gypsum composites. The composites' density, compressive strength, thermal conductivity, and sound absorption were characterized. Results indicate that a composite with a compressive strength from 15 to 136 kPa and a material density ranging from 48 to 194 kg/m$^3$ can be obtained. Thermal conductivity was achieved from 0.0390 to 0.0604 W/(mK). Following the ISO 10534-2 standard, the noise reduction coefficient was determined and showed promising results at 600 to 800 Hz, reaching a sound absorption coefficient of 0.88.

**Keywords:** waste recycling; composite materials; thermal insulation; acoustic materials; EPS aggregate; gypsum

## 1. Introduction

Every year, the amount of construction and demolition waste (CDW) keeps increasing, which is a big problem. In 2018, in Europe, the total share of CDW waste out of all waste was 36%. In some countries, this share is increased to around 70% [1]. Landfills are becoming overfilled with potentially usable material. This can be changed by recycling the waste into new, secondary raw materials used to make new building materials. Nowadays, plastic materials are often labeled on packaging as "made with recycled material". For example, one of the biggest soft drink manufacturers, Coca-Cola, sources 7% of its plastic from recycled materials [2]. This policy should be implemented in the building material sector as it would lower the material costs and be more eco-friendly. However, there is a problem. Using secondary raw materials can worsen the material's properties and quality. In order to avoid this, new methods and technologies must be developed that improve the properties of secondary raw materials and let the secondary materials compete with virgin materials in the building material market [3].

One such material that increases in volume each year is expanded polystyrene (EPS). It has been the most used material for insulation because of its versatility, light weight and reliability for over 50 years [4]. EPS as a material is 100% recyclable, and its properties after recycling stay the same as those of virgin EPS [5]. The processing can be implemented using five different methods—mechanically, by dissolution, chemically, through incineration with energy recovery, and, which is the least eco–friendly option, by disposal through landfills. Dilution and chemical recycling are not as developed as incineration and mechanical

recycling. A study on Europe's EPS waste found that in 2018, only 10% (14 kt) of all EPS construction waste was recycled, 67% (95 kt) was incinerated, and a staggering 33 kt or 23% were landfilled [6]. In recent years, few legislations have been introduced banning EPS from landfills, which means that more recycling methods must be developed.

Expanded polystyrene (EPS) provides an effective thermal barrier. However, it is not an adequate sound barrier when used alone. Even though it has sound absorption characteristics (see Table 1), it lacks sufficient density, making it inefficient as an acoustic panel to mount on a wall [7]. When used in combination with other materials, however, EPS may be a highly effective sound barrier [8,9]. It has a variety of properties that make it excellent in preventing sound transmission when used in wall building.

**Table 1.** Materials for noise control [7].

| Category | Description | Purpose | Representative Uses |
|---|---|---|---|
| Absorptive materials | Relatively lightweight; porous, with interconnecting passages; poor barrier | Dissipation of acoustic energy through conversion to minute amounts of heat | Reduction in reverberant sound energy; dissipation of acoustic energy in silencers |
| Silencers | Series or parallel combination of reactive elements | Dissipation of acoustic energy in the presence of a steady flow | Duct silencers in inlet and exhaust silencers for engines, fans, turbines |
| Barrier materials | Relatively dense, nonporous | Attenuation of acoustic energy | Containment of sound |
| Damping treatments | Viscoelastic materials with relatively internal losses | Dissipation of vibratory energy | Reduction in acoustic energy |
| Vibration isolators | Resilient pads; metallic springs | Reduction in transmitted forces | Mounts for fans, engines, machinery |

For the last ten years, researchers have studied [10–14] the different composites containing EPS for applying thermal and acoustic properties, but they were not feasible for industrial applications. Karina A. de Oliveira studied the casting method to obtain closed-structure gypsum composites with EPS and cellulosic pulp. That resulted in a dense material (370–510 $kg/m^3$) with low acoustic properties because of the large density [13]. A composite made with a semi-dry method could provide better acoustic and thermal properties and lower overall density.

Gypsum is a widely used binder in the construction sector and has a wide range of applications, from plasterboards to monolith casting [15,16]. Gypsum, compared with cement, has low energy consumption and can be reasonably easy to recycle [17,18]. Low emissions for gypsum production show that it is a sustainable material and can close the life-cycle loop and become an eco-friendly building material [19,20]. Using gypsum as a binder with different waste materials results in more robust, eco-friendly, sustainable materials with various properties that can be changed for the necessary application [21,22].

This study aims to make a low-energy production composite material containing thermally modified waste EPS aggregates and gypsum as a binder. Open-structure gypsum composite was developed and investigated for the mechanical, thermal and acoustic properties in self-bearing building envelope systems.

## 2. Materials and Methods

### 2.1. Materials

In this study, raw materials are waste-expanded polystyrene aggregates such as aggregates and commercially available gypsum (CG). CG set time was $t_{st}$ 18:30 min and $t_{fin}$ 22:50 min. Granulometry showed that $d_{10}$ = 0.08 mm, $d_{50}$ = 0.13 mm and $d_{90}$ = 0.22 mm, and the total amount of gypsum was 93%. From the available technical specification of CG, the mixture composition of CG was as follows: $SiO_2$—3.73%, $Al_2O_3$—1.68%, $Fe_2O_3$—0.46%, CaO—35.64%, MgO—3.92%, $SO_3$—30.90%, $Na_2O$—0.31%, $TiO_2$—0.05%, LOI—22.43%.

Recycled EPS aggregates (EPS10) were acquired from a local distributor; the local distributor acquired them from a distribution centre in Poland. The aggregates were packed in 200 l sacks with an average weight of 2 kg. The granule mixture had different aggregates (grey, black, white, red, pink, blue, and green). Most aggregates were individual and wholly separated, although a few remained clustered. The size of these granule clusters varied from 5 to 11 mm.

The physical properties (Table 2) and appearance of raw EPS aggregates were evaluated. The bulk and material density of EPS aggregates, particle size distribution, and pore structure were also determined. EPS aggregates were denoted as EPS10 based on their bulk density.

**Table 2.** EPS10 physical properties.

| Raw EPS | Value |
|---|---|
| Thermal conductivity, W/(mK) | 0.0410 |
| Bulk density, kg/m$^3$ | 10.56 |
| Aggregate density, kg/m$^3$ | |
| 2–4 mm | 21.4 |
| 4–5.6 mm | 26.9 |
| 5.6–8 mm | 11.6 |
| 8–11.2 mm | 16.3 |

Figure 1 shows the particle size distribution of the modified EPS aggregates. All EPS10 aggregates were under 11.2 mm in diameter, with 8–11.2 mm aggregates accounting for 0.71% of all aggregates. Fractions of 5.6–8 mm account for 21.28% of all aggregates, while fractions of 4–5.6 mm account for 52.14%. Particles with a 2–4 mm diameter account for 21.89% of all aggregates, whereas those with a diameter of 1–2 mm and 0–1 mm account for 1.63% and 2.34%, respectively. EPS10 aggregates had a bulk density of 10.56 kg/m$^3$. The following thermal conductivity was achieved: 0.041 W/(mK). The density of the individual aggregates ranged from 11.6 to 26.9 kg/m$^3$. Thermal treatment at elevated temperatures gradually reduced granule dimensions while increasing bulk density.

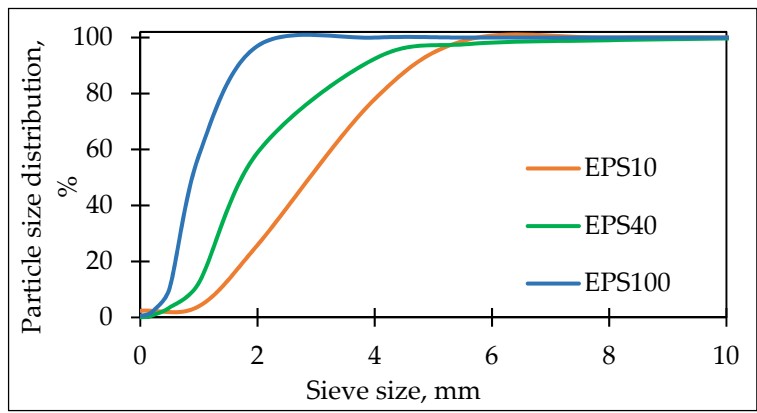

**Figure 1.** Particle size distribution of waste and modified EPS aggregates.

Based on this study [5], the EPS aggregates were similarly modified. The aggregates were modified with different modification times and temperatures to repeat the mentioned study and obtain results with different times and temperatures to compare which would be the most feasible for this study. The modification was performed in a curing chamber with a turned-off fan to keep aggregates stationary, as they are light and would fly everywhere in the curing chamber. The two main factors of successful modification are aggregate visual look and bulk density. Overall modification of EPS aggregates was conducted in two stages.

In the first modification stage, two modification times, 7.5 and 15 min, were selected based on the previous study (see Table 3). The temperature was 110 °C to 150 °C with a

step of 10 °C to obtain more pronounced data on the overall effect on the EPS aggregates by this modification process. By doing this, a trendline can be made of this process that can be adapted to the following stages of the modification. Data gathered in the first stage of the modification can be seen in Table 2. Data for 15 min treatment at 150 °C were not included as the aggregates had melted and were unusable for further study.

**Table 3.** Bulk density of the first stage of EPS aggregate modification.

| Temperature of Modification, °C | Modification Time, Minutes | Bulk Density, kg/m³ |
|---|---|---|
| 110 | 7.5 | 9.6 |
|  | 15 | 10.6 |
| 120 | 7.5 | 11.0 |
|  | 15 | 21.0 |
| 130 | 7.5 | 9.0 |
|  | 15 | 24.6 |
| 140 | 7.5 | 10.1 |
|  | 15 | 62.2 |
| 150 | 7.5 | 25.4 |

The second stage consisted of modification at 120 °C and 130 °C, with the modification time of 7.5; 15; 20; 30; 40; 45 min to acquire better results in these specific temperatures and achieve the bulk density range using these modification times. Forty-five minutes were chosen as the last modification time based on the gathered data and the feasibility of the aggregates.

Based on Figure 1, all of the EPS40 aggregates were under 8 mm in diameter, and all of the EPS100 aggregates were under 4 mm. The EPS40 aggregates had a bulk density of 43 kg/m³, and the EPS100 aggregates had a bulk density of 105 kg/m³. The most significant difference can be observed for particle sizes under 2 mm. EPS10 showed that 26% of all aggregates were under 2 mm, and EPS40 and EPS100 showed 59% and 97%, respectively, which means the heat treatment was successful. Figure 2 represents the bulk density effect on modification time.

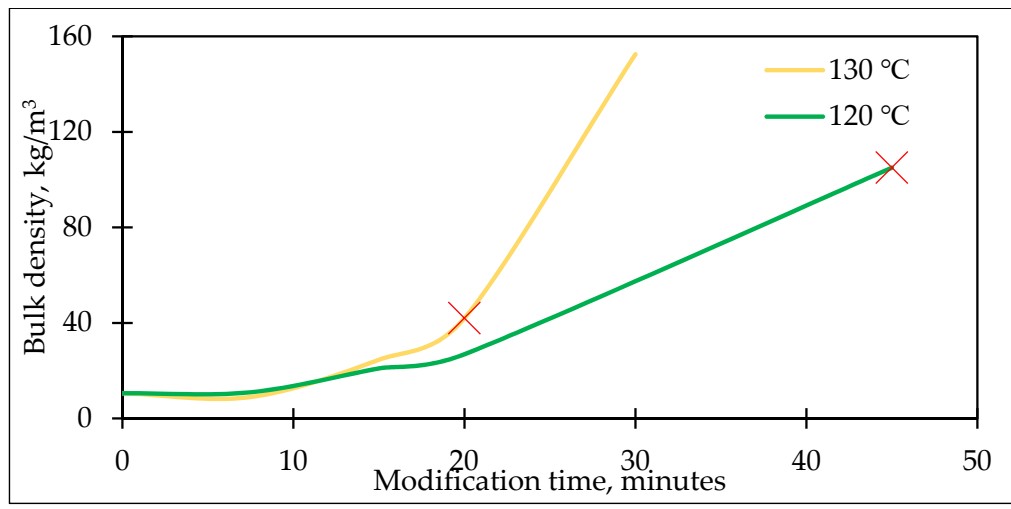

**Figure 2.** Bulk density effect on the modification time of EPS aggregates.

Bulk density increased from 10.6 kg/m³ to 105.6 kg/m³ during thermal treatment at 120 °C from 0 to 45 min. As for the modification at 130 °C, the density increased exponentially from 10.6 kg/m³ to 152.5 kg/m³.

Two modification methods were selected for the mixture design of the gypsum–EPS composite and sample preparation. Modification at 120 °C for 45 min was suffi-

cient to obtain aggregates with a bulk density of 105.6 kg/m$^3$ and thermal conductivity of 0.0380 W/(mK), which were used as EPS100. Modification at 130 °C for 20 min resulted in EPS aggregates with a bulk density of 42.5 kg/m$^3$ and thermal conductivity of 0.0350 W/(mK), which were used as EPS40. The modification was performed a few times to acquire the necessary volume of aggregates for further compositions. Figure 3 depicts the appearance and macrostructure of EPS aggregates of various particle sizes. The most prominent EPS aggregates have a spherical shape with angular planes, indicating that they were previously compressed as packaging material. After manufacture, raw EPS aggregates have a spherical shape which changes after heat treatment during the formation of insulating packaging plates. Small particles with cut sides are most commonly formed during the grinding of recycled EPS plates (Figure 3c).

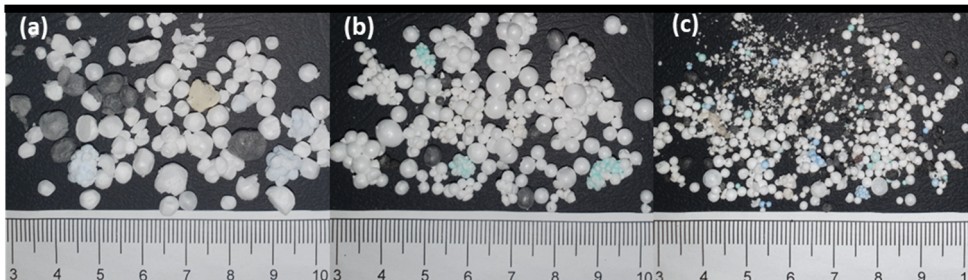

**Figure 3.** Appearance and macrostructure of different-particle-size waste and modified EPS aggregates. (**a**) Raw waste EPS; (**b**) modified at 130 °C; (**c**) modified at 120 °C.

### 2.2. Mixture Design and Sample Preparation

For this study, a batch-type production method of using pressure was used (Figure 4). This resulted in a more compacted composite, where EPS aggregates are closer, ensuring less gypsum is needed to fill the voids between the aggregates. Additional reduction in gypsum can be achieved by improved bonding provided by applied pressure. Compared to the traditional casting method, the advantage of using this compression method is the preparation of thermal insulation materials containing EPS aggregates. Slight deformation of EPS aggregates might occur in the compression process due to flexibility (compressibility), which is also an essential reason why the samples can be compressed. Samples were prepared with added pressure. Lower pressure increases the proportion of EPS aggregates, leading to decreased density and improved thermal insulation performance.

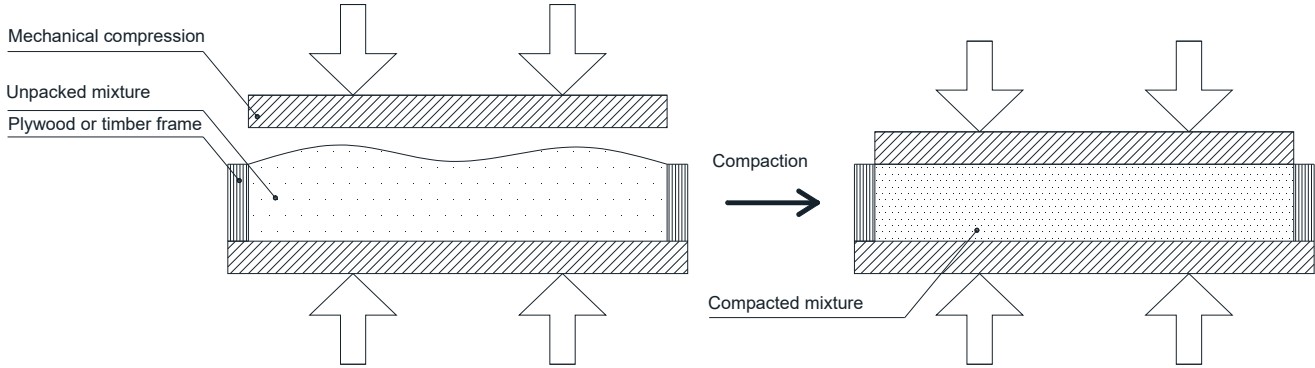

**Figure 4.** The batch-type production method of semi-dry gypsum EPS composite with pressure application [23].

Curing under pressure differs from the casting method, mainly because of the added weight to sustain the different pressure rates. The casting method is traditionally associated with high density, while the semi-dry method can reduce the material's density, which is this study's aim.

The compression ratio was the ratio of the volume before and after compression. The compression ratio of the overall samples was about 10%, which means that the compacted material is 10% smaller in volume than the standard casting method material.

Mixture compositions of prepared EPS–gypsum composites are given in Table 4. Gypsum and water content were changed to evaluate their effect on the composite. Gypsum content was 300 g; 600 g and 1200 g used per sample, and the W/B ratio was reduced from 1.00 to 0.42. Reducing the water-to-binder (W/B) ratio helps decrease the amount of free water present in the composite, which in turn reduces the energy required for drying the sample.

**Table 4.** Mixture compositions of EPS and gypsum samples.

| Series | EPS, g | Gypsum, g | $H_2O$, g | W/B Ratio |
|--------|--------|-----------|-----------|-----------|
| EPS10 | 120 | 300 | 300 | 1 |
| | | 600 | 450 | 0.75 |
| | | 1200 | 500 | 0.42 |
| EPS40 | 450 | 300 | 300 | 1 |
| | | 600 | 450 | 0.75 |
| | | 1200 | 500 | 0.42 |
| EPS100 | 1100 | 300 | 300 | 1 |
| | | 600 | 450 | 0.75 |
| | | 1200 | 500 | 0.42 |

For the semi-dry method, first, all of the water was homogenized with the EPS aggregates so that all the EPS aggregates were covered with water and there was no water buildup in the bottom of the mixer. Commercial gypsum powder was gradually added to the mixture as it was being mixed. Mixing occurred until all the gypsum was added and the final mixture was homogeneous. The mixture was poured into the plywood form with dimensions of 35 cm × 35 cm × 10 cm. A smaller plywood plate was inserted on top of the material to compress it, and a pressure of 1.68 kPa was applied with a weight of 21 kg.

*2.3. Characterization Techniques*

2.3.1. Scanning Electron Microscopy (SEM)

The Hitachi TM3000 tabletop scanning electron microscope (SEM) was used to analyze the microstructure and morphology of the samples. SEM provides high-resolution imaging of surface structures and features, and the Hitachi TM3000 can image samples at magnifications of up to 50,000×. Its low vacuum mode enables imaging of non-conductive samples without needing a conductive coating. The microscope's large chamber accommodates samples of up to 70 mm in diameter and 50 mm in height and includes a variety of detectors for imaging and analysis. SEM analysis provided valuable insights into the microstructural properties of the samples.

2.3.2. Thermal Conductivity

In compliance with EN 12667, the coefficient of thermal conductivity was measured using the Laser Comp heat flow measurement device FOX 600 (Figure 5a). Horizontally fastened samples with dimensions of 350 mm × 350 mm and a 75 to 95 mm thickness were used to determine thermal conductivity.

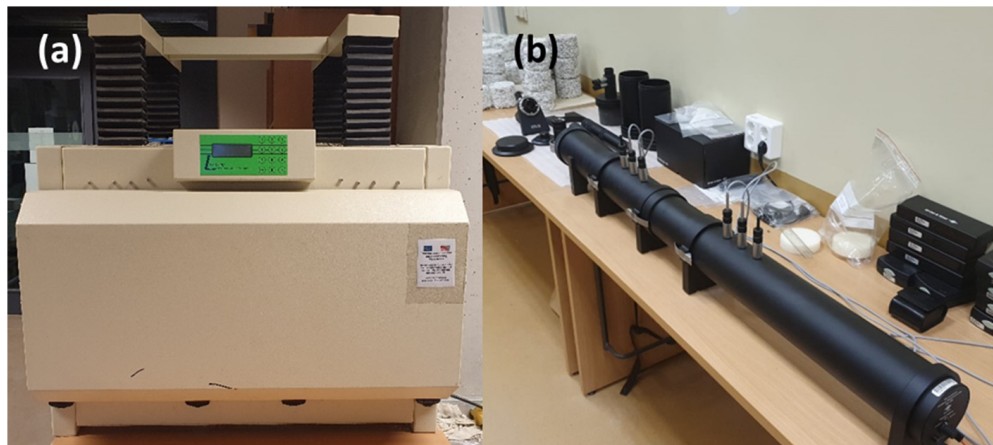

**Figure 5.** Thermal conductivity coefficient testing apparatus (**a**) and sound absorption coefficient testing equipment (**b**).

The lower plate of the device had a temperature of +20 °C, whereas the upper plate had temperature of 0 °C. The heat transmission coefficient was calculated using a continuous heat flow between the two surfaces of the samples.

Thermal conductivity is shown as an upper and lower limit. For the data, an average of these two values was used.

### 2.3.3. Compressive Strength

The compressive strength of cubic specimens was measured with a Zwick Z100 universal testing device at a testing speed of 0.5 mm/min. Before crushing, the cubic specimens were also measured and weighted to measure the material density and volume. Density was calculated by dividing the mass of the sample by the volume of the sample. The force applied to the sample's specific surface area was used to compute the compressive strength.

### 2.3.4. Sound Absorption

Noise Reduction Coefficient (NRC) and Sound Transmission Class (STC) measurements determine how effectively a material absorbs or blocks sound. NRC was measured on a scale of 0.0 to 1.0. The average noise control between the frequencies of 50 and 1600 Hz is shown in this test: the density, thickness, and overall material impact the ability to absorb or block sound. The ISO 10534-2 standard [24] uses the impedance tube method to calculate the sound absorption coefficient. The test is performed after a cylindrical sample is placed into the tube and the tube is closed (Figure 5b). This is not a long-waiting test; one sample can collect results in half a minute to a minute.

## 3. Results and Discussion

### 3.1. Microscopy of EPS Composites

The visual appearance of different composites is compiled in Figure 6. All images are magnified 25-fold to analyze their differences by varying the gypsum amount and modification temperature effect on aggregates. In the first row of pictures, EPS10 granules have been compiled with increasing gypsum amounts from left to right. The second row is EPS40 composites, and the third is EPS100, respectively. The gypsum amount is compiled in the columns, with increasing gypsum from left to right. Analyzing the images by rows, composite aggregates can be seen to shrink. In the first row, barely two aggregates can be seen, and after the application of different modifications at previously selected elevated temperatures, the aggregates change their structure, becoming denser and smaller in size. In the third row, where EPS100 composites are visualized, aggregates can be fully seen, which confirms the hypothesis of aggregate size change based on the modification.

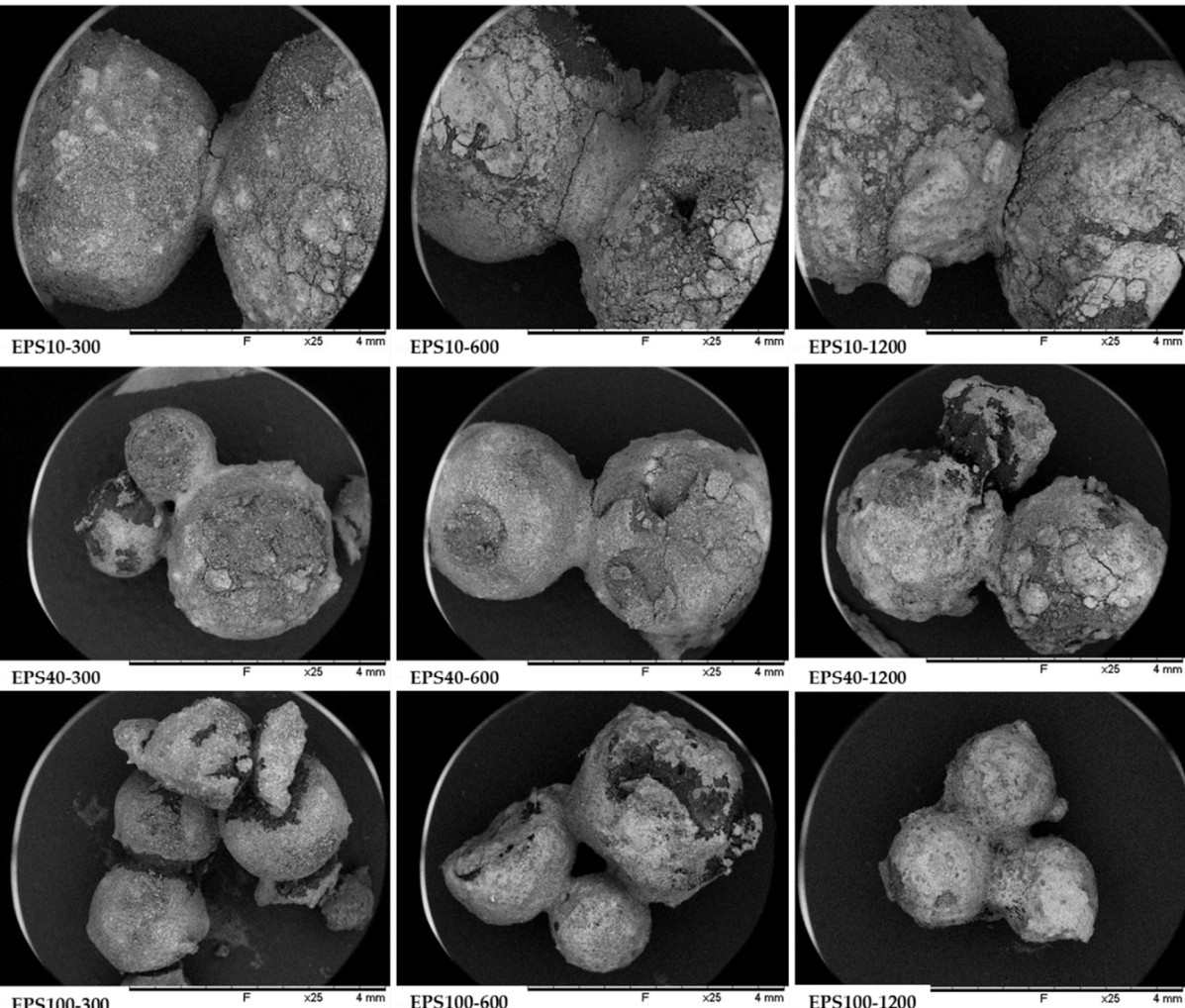

**Figure 6.** Macroscopic SEM imaging of the study's composites at a 25-fold magnification.

The effect of increased gypsum can be observed by increasing the magnification 50-fold (Figure 7). By increasing the added gypsum, a denser layer of gypsum can be seen on the aggregates and around the contact zones. In the left column, where the smallest amount of gypsum was added, a thin layer of gypsum can be seen, with some of the spaces not filled, resulting in black spots in SEM images. Black spots are the EPS polymer that is visible through gypsum particles. After the increase in the gypsum amount, a more stable coating of EPS aggregate can be observed. Almost all of the surfaces of the aggregates are coated with gypsum, and the contact zones, where two aggregates connect, are also bound together with a dense layer of gypsum. Further increasing the gypsum amount results in an aggregate with a very thick layer of gypsum which forms an uneven surface and a dense contact zone, which even covers the middle part of the three aggregates seen in the bottom right image. The reduction in open porosity between individual EPS aggregates contributes to the physical and mechanical properties described in the following paragraphs.

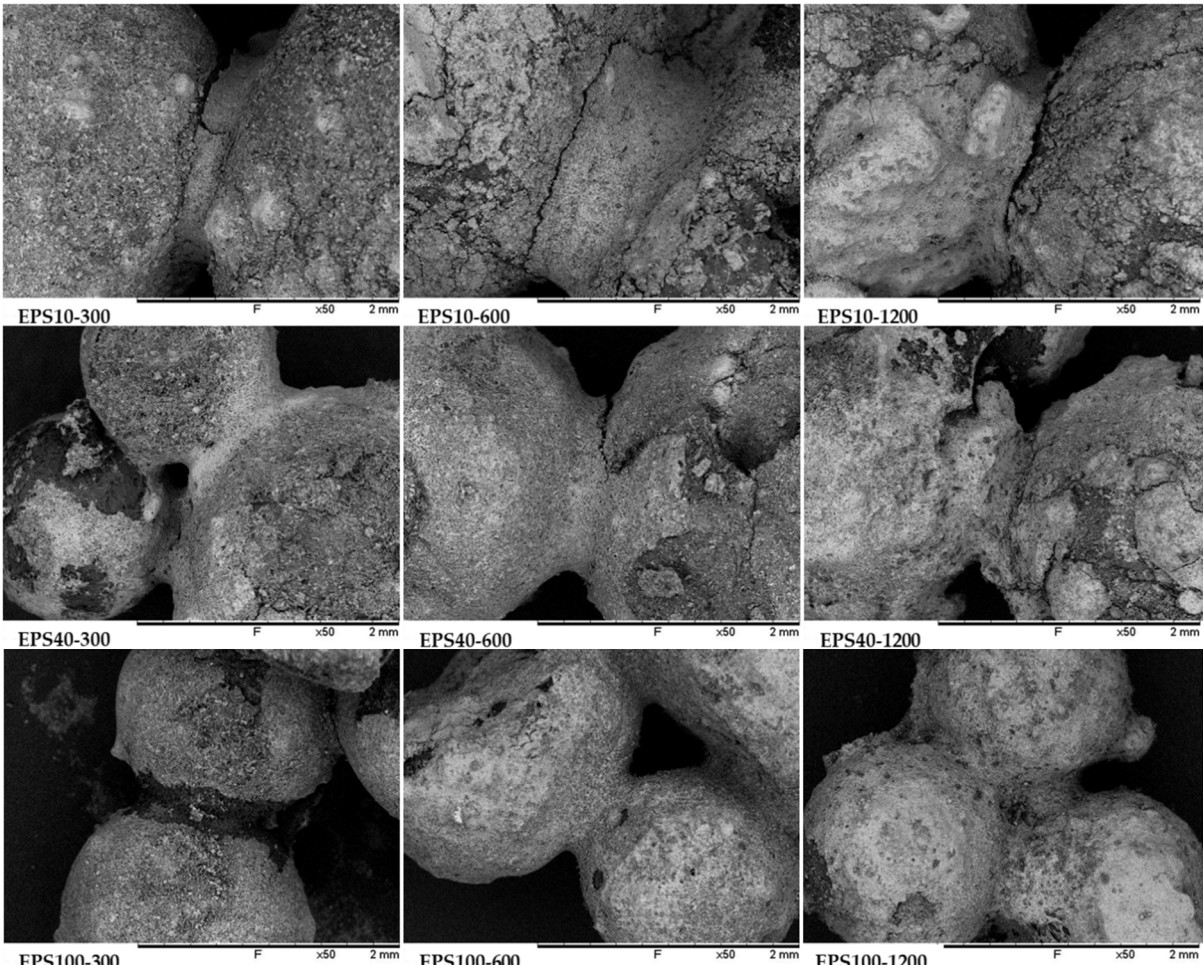

**Figure 7.** SEM imaging of the study's composites at a 50-fold magnification.

Further increasing the magnification to 100-fold, an analysis of the contact zones can be established. Looking at Figure 8, fractures can be observed in all of the EPS10 composites. As the aggregate is easily deformed, the gypsum layer breaks and shows fractures. These fractures can decrease the composites' mechanical properties as the deformed bounded region. By increasing the bulk density of the aggregates to 40 kg/m$^3$ (EPS40), a decrease in fractures can be noticed, resulting in a more stable composite. As the bulk density increases, the size and deformability decrease, resulting in a better bond between the aggregates. No fractures can be observed by increasing the bulk density of the aggregates to 100 kg/m$^3$ (EPS100).

The increase in gypsum content in EPS composites made obtained composite less brittle. Loose structures with weak bonding between individual EPS aggregates are associated with low gypsum content. The application of such composite has limited handling possibilities as cutting and processing of such material is problematic. The thermal treatment of the EPS aggregate produced a higher specific surface area of aggregates than the untreated EPS. Therefore, the gypsum content of both compositions reduced the interaction zone for thermally treated EPS granules. Solid EPS composite was achieved through the highest gypsum content evaluated in this research. Increased gypsum content ensured the interlocking of EPS granules, making the material more suitable for processing.

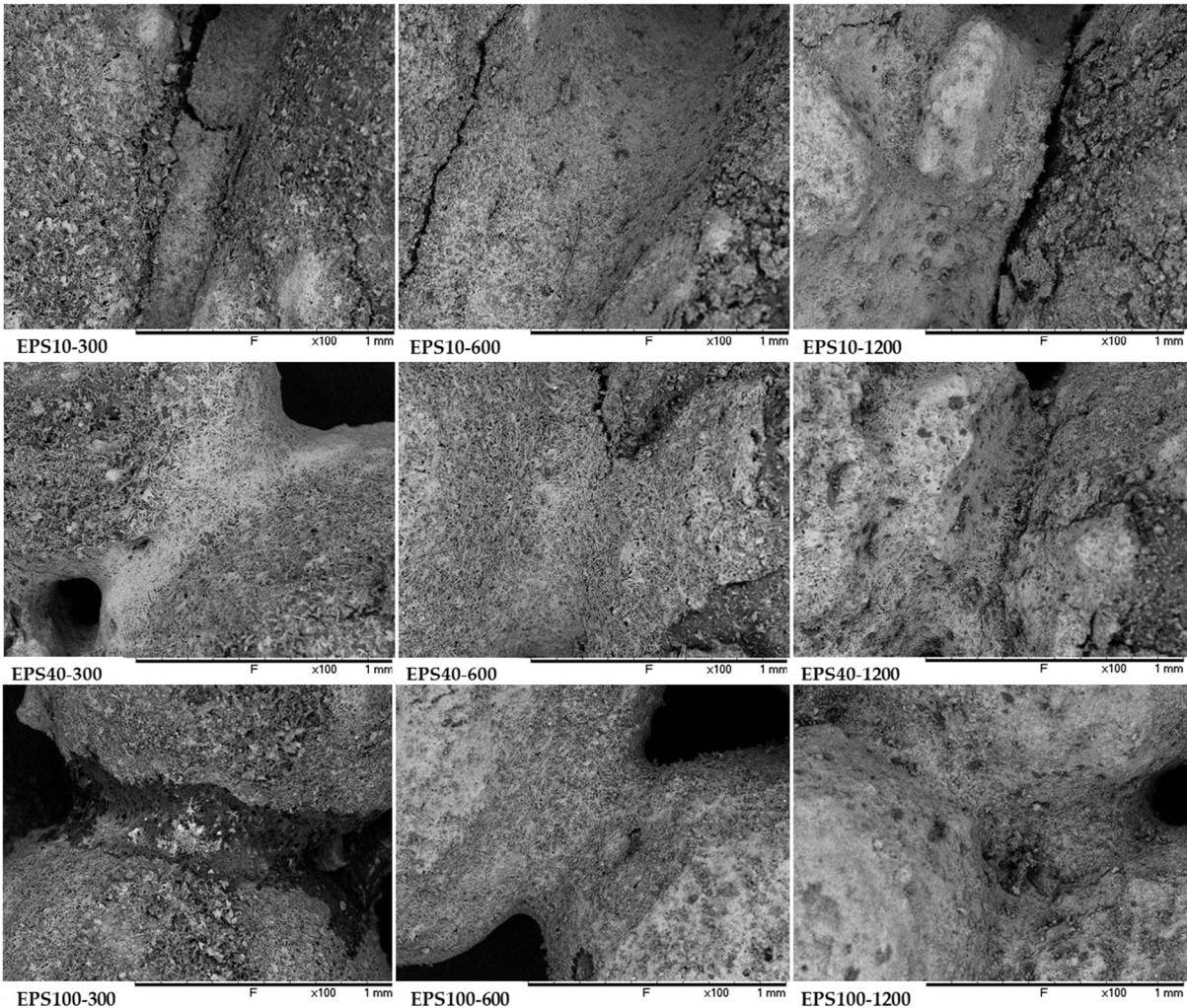

**Figure 8.** SEM imaging of the study's composites at a 100-fold magnification.

### 3.2. Properties of Gypsum and EPS Composites

Gypsum content had a significant role in the material density of the composites. After increasing the amount of gypsum, an increase in density, compressive strength and thermal conductivity can be observed. Table 5 comprises the physical properties of gypsum and EPS composites.

**Table 5.** Physical properties of the EPS and gypsum composites.

| Composition | | Material Density, kg/m³ | Thermal Conductivity, W/(mK) | Compressive Strength, kPa |
|---|---|---|---|---|
| EPS10 | | 48 ± 1 | 0.0390 | 21 ± 1 |
| EPS40 | 300 | 73 ± 3 | 0.0396 | 18 ± 6 |
| EPS100 | | 122 ± 10 | 0.0426 | 15 ± 8 |
| EPS10 | | 74 ± 1 | 0.0458 | 29 ± 4 |
| EPS40 | 600 | 93 ± 2 | 0.0445 | 61 ± 5 |
| EPS100 | | 136 ± 3 | 0.0462 | 46 ± 7 |
| EPS10 | | 154 ± 2 | 0.0565 | 50 ± 7 |
| EPS40 | 1200 | 177 ± 6 | 0.0604 | 136 ± 12 |
| EPS100 | | 194 ± 5 | 0.0558 | 115 ± 26 |

The data suggest that as the bulk density of EPS increases, the material density also increases. This trend is observed across all three gypsum series for each type of EPS. For

example, EPS10 has a lower material density than EPS40, which in turn has a lower material density than EPS100. Material density increases with the added gypsum and the increased bulk density of the modified EPS aggregates. EPS10-300 achieved the lowest material density—48 kg/m$^3$. By increasing the gypsum amount to 600 and 1200, a material density of 74 (EPS10-600) and 154 kg/m$^3$ (EPS10-1200) was obtained. If we compare the density differences based on the increase in aggregate bulk density, a similar trend can be noted. EPS10-600 achieved 74 kg/m$^3$, EPS40-600 and EPS100-600—93 and 136 kg/m$^3$, respectively. The largest material density was obtained for the sample with the largest added gypsum and the highest aggregate bulk density (EPS100-1200), which was 194 kg/m$^3$.

Looking at Figures 9 and 10, a relationship between compressive strength, thermal conductivity, and material density can be analyzed. A comparison with material density was made to analyze the industrial application of material manufacturing. Samples showed compressive strength ranging from 15 to 136 kPa, with the material density ranging from 48 to 194 kg/m$^3$. Thermal conductivity was achieved from 0.0390 to 0.0604 W/(mK).

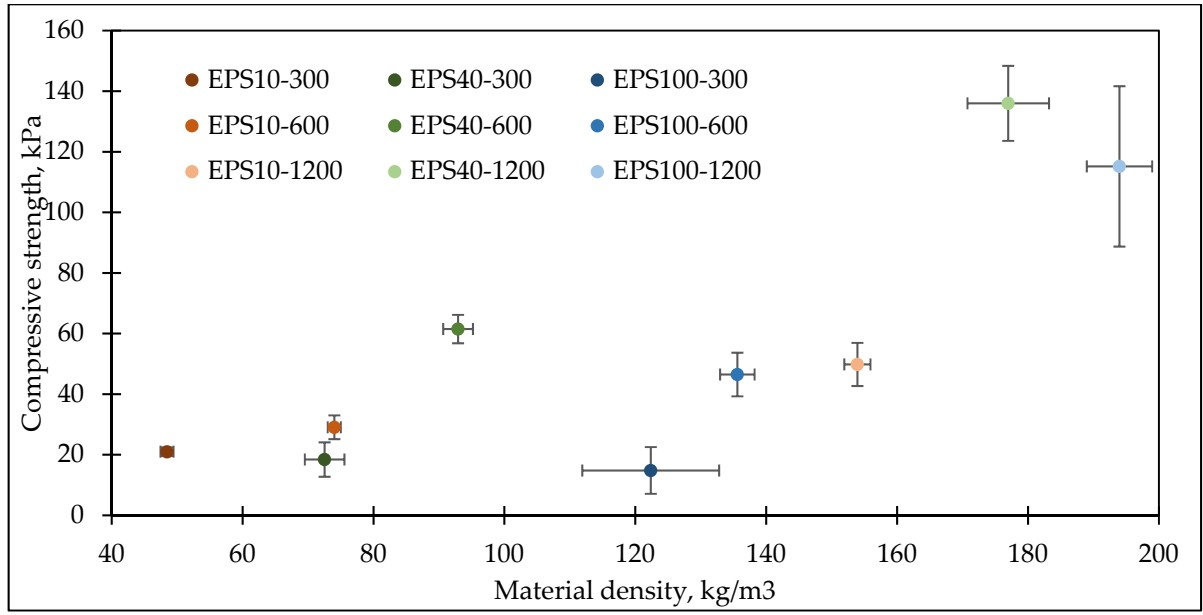

**Figure 9.** Correlation between compressive strength and material density.

The highest compressive strength of the samples series with 300 g gypsum was the one with non-modified EPS aggregates, resulting in compressive strength of 21 kPa. The second highest compressive strength was the next sample series that used the modified EPS aggregates with a bulk density of 40 kg/m$^3$ (see Section 2.1) and resulted in 18 kPa. The lowest compressive strength was observed in the modified EPS aggregates with a bulk density of 100 kg/m$^3$ (EPS100). This series showed 15 kPa. By inclusion of marginal error, the samples of 300 g gypsum did not differ much regarding compressive strength.

Increasing the gypsum mass to 600 g, more dense samples were acquired with higher compressive strength. Unlike the first series, this series' highest compressive strength was observed in the modified EPS aggregates with a bulk density of 40 kg/m$^3$ (later—EPS40). The lowest acquired compressive strength from this series was observed in the raw EPS aggregates. The results were 61 and 29 kPa, respectively.

Using 1200 g of gypsum, the densest and highest-compressive-strength materials in this study were obtained. Similar to the previous series, the highest compressive strength was observed in the EPS40 aggregates. The lowest compressive strength was observed in the non-modified EPS aggregates with the amount of 136 and 50 kPa, respectively.

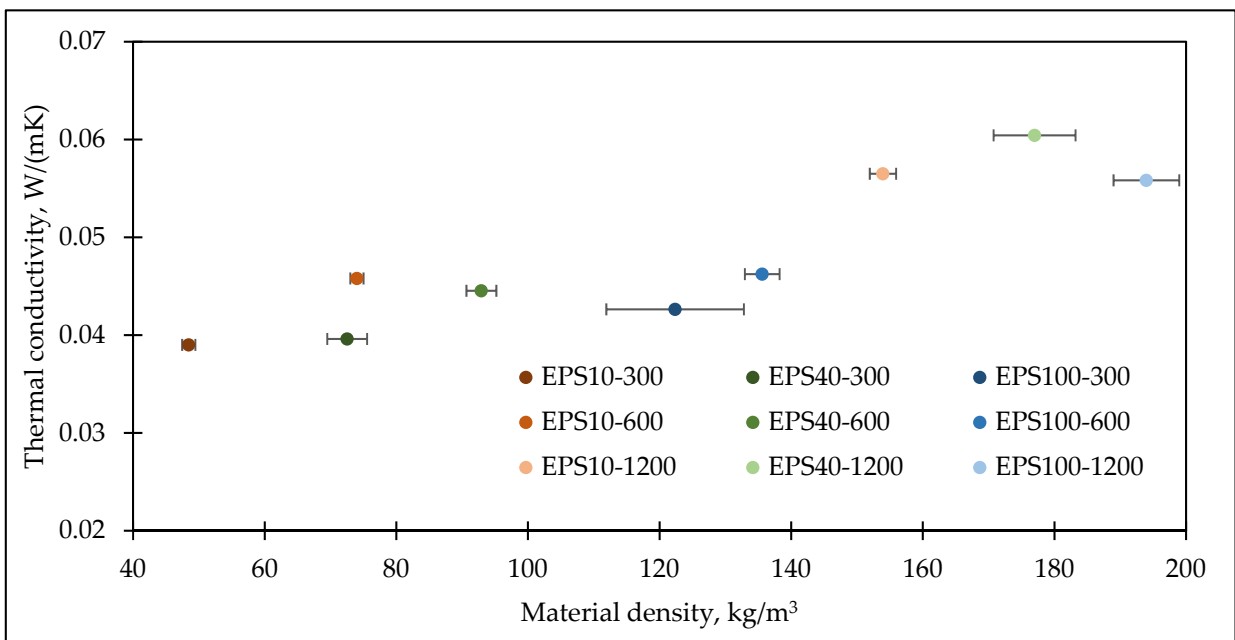

**Figure 10.** Correlation between thermal conductivity and material density.

It is clear from Figure 10 that after an increase in the gypsum mass, the material density and the thermal conductivity increase as well. Further analyzing the figure shows that the modification of EPS aggregates does not affect the thermal conductivity of the overall material comparing the use of the EPS10 and EPS100 aggregates. The results of the EPS40 aggregates results vary from those of the used gypsum mass. In the case of 300 g, thermal conductivity is almost the same as that of the non-modified EPS samples, 0.0396 W/(mK) and 0.0390 W/(mK), respectively. Analyzing the 600 g gypsum series, the EPS40 aggregates showed the lowest thermal conductivity of that series—0.0445 W/(mK). The 1200 g gypsum series showed that using EPS40 aggregates increases the thermal conductivity to 0.0604 W/(mK). Unlike the other EPS aggregates, EPS10 showed 0.0565 W/(mK) and EPS100—0.0558 W/(mK).

*3.3. Sound Absorption of the Composites*

Figure 11 comprises sound absorption for all three sample series varying by the material density of EPS used. The EPS10 sample series represents sound absorption for non-modified EPS aggregate samples; a sound absorption of over 0.7 can be achieved in the 550 to 875 Hz frequency range. The maximum sound absorption of the EPS10-300 was 0.82 at 735 Hz. The other two samples showed slightly different sound absorption. EPS10-600 achieved 0.88, and EPS10-1200 achieved 0.86 at 675 Hz.

EPS40-300 showed sound absorption of 0.7 in the 765–900 Hz range with a peak of 0.74 at 808 Hz. EPS40-600 and EPS40-1200 showed similarities reaching the threshold at 620–873 and 629–911 Hz, respectively. The series also showed an increase in the maximum sound absorption value, reaching 0.84 at 718 and 762 Hz, respectively.

EPS100-300 achieved sound absorption of 0.7 in the 680 to 855 Hz frequency range with two different maximum peaks of 0.74 and 0.75 at 763 and 805 Hz, respectively. EPS100-600 achieved broader absorption above the 0.7 threshold, 531–821 Hz, with a similar maximum double peak achieving 0.88 and 0.87 absorption at 572 and 665 Hz, respectively. EPS100-1200 achieved the broadest range above the 0.7 threshold in the 525 to 852 Hz range with a maximum peak of 0.88 at 687 Hz.

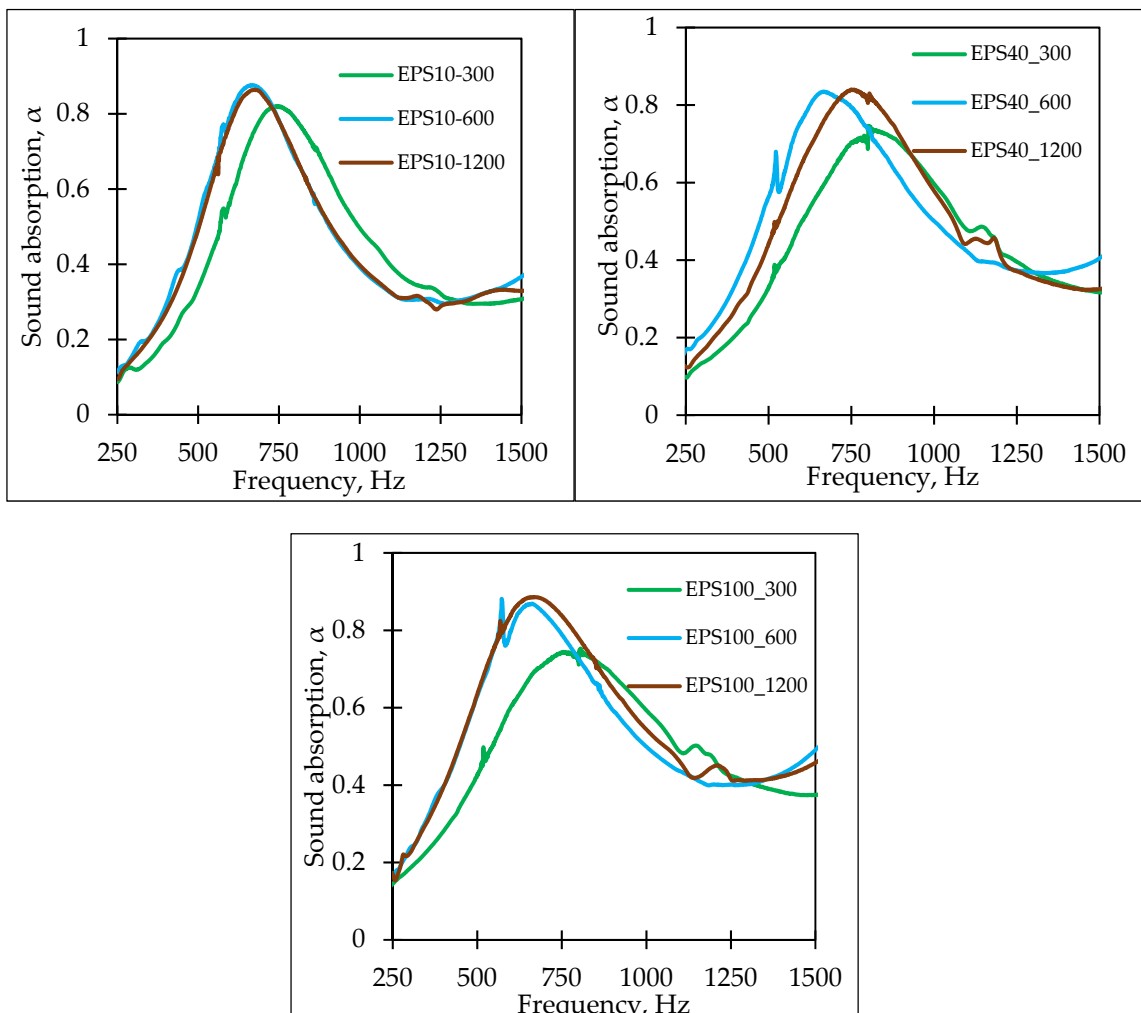

**Figure 11.** Sound absorption of EPS10, EPS40 and EPS100 samples.

## 4. Conclusions

The study explores the properties of expanded polystyrene (EPS) aggregates and how they can be modified to enhance their characteristics. The study seeks to understand the effects of EPS aggregate density, temperature deformation point, and thermal conductivity through particle size distribution data and thermal treatment. Additionally, the study investigates how modifying EPS aggregates with gypsum can improve their compressive strength and sound absorption, and the following conclusions can be drawn:

- Based on the presented granulometry data, it can be concluded that thermal treatment affects EPS aggregates by reducing volume and thus increasing bulk density;
- EPS aggregates reach a melting point if the thermal treatment temperature is set to 150 °C for longer than 7.5 min;
- A total of 97% of all EPS100 aggregates are under 2 mm, while 26% and 59% of EPS10 and EPS40 are under 2 mm, respectively;
- It has been determined that bulk density can be increased 10-fold by thermal treatment at 120 °C, and by 4-fold at 130 °C treatment;
- After an increase in the gypsum mass, the compressive strength and material density increase as well, with compressive strength ranging from 15 to 136 kPa and the material density ranging from 48 to 194 kg/m$^3$;
- The modification of EPS aggregates does not affect the thermal conductivity of the overall material when comparing the use of EPS10 and EPS100 aggregates;

- The lowest thermal conductivity was found to occur for samples made with 300 g of gypsum, 0.0390 and 0.0396 W/(mK) using EPS10 and EPS40 aggregates, respectively;
- The highest sound absorption was achieved by samples EPS10-600, EPS100-600 and EPS100-1200 reaching 0.88 at 675, 572 and 687 Hz.

In summary, this study provides valuable insights into the properties of EPS aggregates and their modification with gypsum. The particle size distribution data and thermal treatment analysis demonstrate that thermal treatment can significantly affect the bulk density of EPS aggregates, with higher temperatures leading to increased density. Additionally, modifying EPS aggregates with gypsum is shown to improve their compressive strength and sound absorption. Notably, the thermal conductivity of the overall material is affected slightly by the modification of the EPS aggregates. These findings can be helpful for engineers and designers who are interested in developing materials with enhanced characteristics for various applications such as thermal insulation and soundproofing.

**Author Contributions:** Conceptualization, P.P.A., G.B. and D.B.; methodology, P.P.A., G.B. and D.B.; software, P.P.A., G.B. and D.B.; validation, P.P.A. and G.B.; formal analysis, P.P.A. and G.B.; investigation, P.P.A., G.B. and D.B.; resources, D.B.; data curation, P.P.A. and G.B.; writing—original draft preparation, P.P.A.; writing—review and editing, P.P.A. and G.B.; visualization, P.P.A.; supervision, G.B. and D.B.; project administration, D.B.; funding acquisition, D.B. All authors have read and agreed to the published version of the manuscript.

**Funding:** This research was funded by the FLPP (Fundamental and Applied Research Projects) Pro-gram in Latvia under the research project lzp-2020/1-0010, "Reuse of gypsum and expanded polymers from construction and demolition waste for acoustic and thermal insulation panels.".

**Data Availability Statement:** Data are available on request due to project privacy restrictions.

**Conflicts of Interest:** The authors declare no conflict of interest.

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
