# Peer review of "Gypsum Composites with Modified Waste Expanded Polystyrene"

_jcs, doi:10.3390/jcs7050203_

Round 1

Reviewer 1 Report

This is an excellent manuscript. Below a few comments to further improve this work:

-line 39-49: So why is it not done yet? Too expensive?

-line 64-70: gypsum seem to actually also become a higher demand commodity (https://doi.org/10.1016/j.resconrec.2022.106328). Maybe use phosgypsum as the authors suggest. It seems that this could be a good outlook for your work since there is soo much near Kaunas, not too far away from your country…

-please specify the method of analysis

-the experiments were conducted well, and you also provide excellent explanation on all this in the manuscript. I think the economic point of the process can be discussed, since this is probably what holds back large scale application…

Author Response

The authors have reviewed the comments and have answered in the attachment.

Reviewer 2 Report

The paper is structured properly (INTRO, MATERIAL & METHODS, RESULTS & DISCUSSION, CONCLUSIONS and REFERENCES).

The INTRODUCTION section provide the necessary background information regarding the fluorine materials and the research on material performance prediction based on machine learning. The section is well–documented. Also, the aims and objectives of the research are well defined.

The METHODOLOGY / MATERIALS & METHODS section is relatively well described and include detailed information. The body of paper describe the important RESULTS of the research, followed by several DISCUSSIONS.

The CONCLUSION section succinctly summarize the major points of the paper, derived from the results and the discussions. The authors fairly concludes in just a few sentences given the rich discussion in the body of the paper.  It is quite succinctly presented. Are only technical comments, very briefly presented, without other opinions, conclusions or remarks. Indicate what you have done that is new compared to the already known works.

The list of REFERENCES is long and relatively well chosen. The entire bibliography is current, and modern works are mainly used. Literature review provides comprehensive information about the current state of research.

Author Response

The authors thank the reviewer for their thorough analysis and comments. The study's main aim was to modify the EPS granules and incorporate them in a composite material with gypsum used as a binder. This study concentrates on EPS modification as the variable that changes. The authors have other studies that focus on other elements, like waste management and its barriers and the recyclability of the gypsum binder itself. The end goal would be to make a material made from 100% recycled materials incorporating both waste EPS and waste gypsum from construction and demolition waste. The authors again would like to thank the reviewer for his/her review!

Here are the other studies:

DOI: 10.1007/978-3-030-91261-1_25-1

DOI: 10.3390/recycling7030030

Round 2

Reviewer 1 Report

Comments have been addressed.

Manuscript can be published.

Author Response

The authors would like to thank the reviewer for their comments and remarks. Thank you!

Reviewer 2 Report

All improvements are welcome. I appreciate that the authors have taken into account the recommendations of the reviewers. A better synthesis and compaction of the exposed ideas are insured.

The ABSTRACT section is well–structured. The paper is structured properly (INTRODUCTION, MATERIAL AND METHODS, DISCUSSION, CONCLUSIONS, REFERENCES, etc.). The INTRODUCTION section provide the necessary background information. The METHODOLOGY is relatively well described. The body of paper describe the important RESULTS of the research.

The CONCLUSION section succinctly summarize the major points of the paper, quite ambiguous, but presented concisely and to the point. But, the conclusion is intended to help the reader understand why your research should matter to them after they have finished reading the paper. A conclusion is not merely a summary of your points or a re–statement of your research problem but a synthesis of key points. I would recommend presenting the novelties of this study, the main characteristics that individualize these research. What are the conclusions of the research? What does the author think?

Author Response

The authors have answered the reviewers' comments. See attached file.

Round 3

Reviewer 2 Report

All improvements are welcome. I appreciate that the authors have taken into account the recommendations of the reviewers. A better synthesis and compaction of the exposed ideas are insured.

The list of REFERENCES is long and relatively well chosen. The entire bibliography is current, and modern works are mainly used. Literature review provides comprehensive information about the current state of research.